

# Engineered swift equilibration of brownian particles: consequences of hydrodynamic coupling

Salambô Dago[1], Benjamin Besga[1], Raphaël Mothe[1], David Guéry-Odelin[2], Emmanuel Trizac[3], Artyom Petrosyan[1], Ludovic Bellon[1]* and Sergio Ciliberto[1]

**1** Univ Lyon, Ens de Lyon, Univ Claude Bernard Lyon 1,
CNRS, Laboratoire de Physique, F-69342 Lyon, France
**2** Laboratoire de Collisions Agrégats Réactivité, CNRS, UMR 5589, IRSAMC, France
**3** Université Paris-Saclay, CNRS, LPTMS, 91405, Orsay, France.

* ludovic.bellon@ens-lyon.fr

## Abstract

We present a detailed theoretical and experimental analysis of Engineered Swift Equilibration (ESE) protocols applied to two hydrodynamically coupled colloids in optical traps. The second particle disturbs slightly (10% at most) the response to an ESE compression applied to a single particle. This effect is quantitatively explained by a model of hydrodynamic coupling. Then we design a coupled ESE protocol for the two particles, allowing the perfect control of one target particle while the second is enslaved to the first. The calibration errors and the limitations of the model are finally discussed in detail.

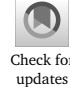
# 1   Introduction

Speeding-up an equilibration process is a delicate task, because the relaxation time is an intrinsic property of a system which depends on parameters such as the dissipation, the potential strength, the inertia, or the number of degrees of freedom. Furthermore, when a control parameter is suddenly changed, the system may pass through states that differ widely from the target one. One way of speeding up a specific transformation between well defined equilibrium states is to apply complex protocols in which the time dependence of one or several control parameters is tuned in a highly specific fashion, to reach the final target in a selected short amount of time. This problem, related to optimal control theory, can be traced back to Boltzmann [1–3]. It has recently received sustained attention within the framework of the so-called "Shortcut To Adiabaticity" protocols, which study such complex procedures for specific transformations  [4,5].

We are interested here in overdamped systems in contact with a thermostat, for which we have defined protocols of Engineered Swift Equilibration (ESE) that have been applied to the control of Brownian particles trapped by optical tweezers [6]. For example, one can achieve the compression of a single particle trapped in an harmonic well by increasing the potential stiffness $K$ between an initial state in equilibrium at $K_i$ and a final state in equilibrium at $K_f$. After a sudden change in $K$ (STEP protocol) the bead will equilibrate in its natural relaxation time. Using an ESE protocol for the time evolution of $K(t)$, the same final state can be reached several orders of magnitude faster than STEP [6]. We will refer to this fast compression protocol as the **basic ESE**. When designing these protocols, one of the key questions lies in the stability against external perturbations. In this context, we tackle in this article the case of two hydro-dynamically coupled particles trapped in different potentials, to understand to what extent the equilibration dynamics imposed by the *basic ESE* is modified by the hydrodynamic interactions with another bead. A deep understanding of the physical consequences of the coupling on the particles behaviour (correlation) is necessary to work out the consequences of this perturbation. The goal here is twofold: on the one hand, it is a simple test bench to probe the robustness of the *basic ESE*. Indeed, we can see the second particle as a perturbation to the first, and monitor how far the protocol misses its target if we neglect this perturbation. And on the other hand, it is a first step towards the control of more complex systems with several degrees of freedom.

The article is organised as follows: in a first part, we investigate robustness of the *basic ESE* to the coupling interaction. To do so, we conduct experiments using the experimental set up described in section 2, and present the results in section 3. To support our experimental results, we then use in section 4 a simple model from refs. [7–10] to describe the coupled system, and predict the dynamics of the correlations at equilibrium and the general dynamics of the moments. We subsequently turn to the second goal of the paper: extending the scope of ESE protocols to more complex systems. The model used is precise enough to provide a basis for the construction of new ESE protocols adapted to the coupled system. In particular we explore in section 5 the construction of ESE protocols that do not depend on the coupling intensity, and are thus very robust. Then we demonstrate experimentally the validity of this extension. Finally we draw the experimental limits of this new strategy in section 6.

# 2   Experimental set up and method

To test the robustness of the *basic ESE* to the coupling interaction, we conduct experiments on two silica beads of radius $r = 1\,\mu$m immersed in miliQ water (to avoid trapping impurities) at a temperature $T$ and trapped by two optical tweezers separated by the distance $d$ (see Fig. 1).

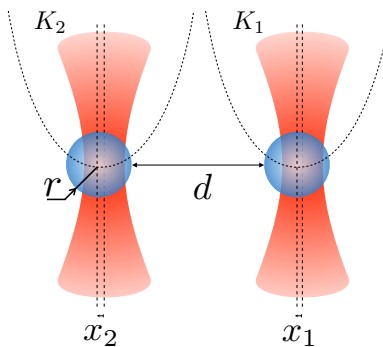

Figure 1: Two Brownian particles trapped by optical tweezers into two harmonic potentials of stiffness $K_1$ and $K_2$. $x_j$ represents the position of the particle $j = 1, 2$ relative to the trap center $x_j^0$, and in the following, $\widetilde{x}_j = x_j + x_j^0$ represents the absolute position. $d$ is the mean distance to contact between the two particles of radius $r = 1\,\mu$m: $d = |x_2^0 - x_1^0| - 2r$.

We use a very small concentration of silica micro-spheres in water and a specific design of the cell containing the particles, in order to have very few beads in the measuring volume. This enables us to take long measurements without any spurious perturbation. The two beads are trapped at $20\,\mu$m from the bottom plate of the cell. The traps are realized using a near-infrared single mode DPSS laser (Laser Quantum, $\lambda = 1064\,$nm used at a power of $1\,$W) expanded and injected through an oil-immersed objective (Leica, $63 \times$ NA 1.40) into the fluid chamber. An Acousto-Optic Deflector (AOD) controls the intensity and the position of the trapping beams with the amplitude and frequency of the control signal, respectively. We thus create two harmonic potentials at a distance $d$ along the $x$ direction $U_j(\widetilde{x}_j, t) = -K_j(t)(\widetilde{x}_j - x_j^0)^2/2$, with $j = 1, 2$, where $\widetilde{x}_j$ are the absolute particle positions. The potential minimum $x_j^0$ and stiffness $K_j$ are controlled respectively by the frequency and amplitude of the AOD input signal. As the AOD responds linearly, a sum of sine waveforms of different frequencies results in two potentials $U_{j=1,2}$ separated by a distance proportional to the difference between the sine frequencies. We can also use a second version of the setup with two AODs (one for each trap) to have two perfectly uncoupled static traps with orthogonally polarized beam (which is needed in particular when $K_1(t) \neq K_2(t)$). The detection of the particle position is performed using a fiber coupled single mode laser diode (Thorlabs, $\lambda = 635\,$nm, power $1\,$mW lowered to $100\,\mu$W with a neutral density filter) which is collimated after the fiber and sent through the trapping objective. The forward-scattered detection beam is collected by a condenser (Leica, NA 0.53), and its back focal-plane field distribution projected onto a four quadrant detector (QPD from First Sensor with a bandpass of $1\,$MHz with custom made electronic) which gives a signal proportional to the particle position. Before every acquisition, a calibration procedure described in Appendix A.1 is conducted.

As regards the acquisition process, the approach consists in comparing the situation when the particles are strongly coupled ($d \lesssim r$), with the situation when the coupling is negligible ($d \gg r$), in order to conclude on the perturbation induced by the coupling. Because the procedure is very sensitive to the instrument calibration and to the external parameters, to compare properly the 2 cases described above, we apply the following protocol: we start at small distance and record the particle position during a dozen of ESE protocols, then we smoothly separate the 2 particles and record again a dozen protocols, before bringing again the 2 particles closer and restart the cycle. Doing so enables us to compare the response to the

ESE protocol in the coupled and uncoupled cases in the same experimental conditions. The recording lasts 10000 protocols to reduce statistical uncertainty. The same approach can be adjusted for other comparisons, the point being always to maintain the same working conditions between the two acquisitions.

## 3 Consequences of coupling perturbation on the *basic ESE protocol*

This section aims to see to what extent the response of the particle to the *basic ESE* deviates from the 0-coupling case successfully tested in ref. [6], when it is affected by the coupling perturbation created by another particle at distance $d$.

Indeed, the *basic ESE* defined in ref. [6] is designed for a single particle trapped in the potential $U(t) = \frac{1}{2}K(t)x^2$, and whose over-damped dynamics is described by a Langevin equation that introduces the friction coefficient $\gamma = 6\pi\eta r$, $\eta$ being the kinetic viscosity and $r = 1\,\mu$m the radius of the particle. The *basic ESE* consists in changing the stiffness over a period of time $t_f$ to reach a new equilibrium at $K_f$. The corresponding stiffness profile is the following, using the dimensionless quantities $k(t) = K(t)/K_i$ (in particular $k_f = K_f/K_i$), $s = t/t_f$ and $\Gamma = \gamma/(K_i t_f)$ (ratio of relevant timescales):

$$k(s) = 1 + (k_f - 1)(3 - 2s)s^2 - \frac{3\Gamma(k_f - 1)(s - 1)s}{1 + (k_f - 1)(3 - 2s)s^2}. \tag{1}$$

One may expect that if the ESE final time $t_f$ is small enough compared to the characteristic correlation time $\tau_{\text{corr}}$, the particles will behave as in the free case. To test this hypothesis we study the evolution of the variance of the first particle during what we call the **symmetric protocol**: the stiffness of both wells is simultaneously driven ($K_1(s) = K_2(s) = K(s)$) according to the *basic ESE* of eq. (1). In what follows we associate with the first particle variance $\langle x_1^2 \rangle$ the dimensionless quantity $\sigma_{11} = K_i\langle x_1^2 \rangle/(2k_B T)$. In the *symmetric protocol* context $\sigma_{11} = \sigma_{22} = \sigma$. We carry out this procedure for an ESE time $t_f$ one order of magnitude smaller than the typical characteristic times $\tau_{\text{corr}} \sim \tau_{\text{relax}} \sim 15\,$ms. To cycle the procedure we use the stiffness profile of Fig. 2 (left) for both traps: a simple step decompression followed by the *basic ESE* compression. The experimental results are plotted in plain lines on Fig. 2 (right), in purple for a small distance and in black for a large distance. Since we look for tiny effects, all results in the article are plotted using the normalised variance $\sigma_n = (\sigma(t) - \sigma_f)/(\sigma_i - \sigma_f)$. In response to the step decompression, the particle reaches equilibrium in its natural relaxation time $\tau_{\text{relax}}$. We notice that the coupling also affects this natural relaxation (by slightly slowing it down). Then we apply the *basic ESE protocol* to both wells, and we observe that at small $d$ the coupling induces a rebound in the variance evolution (indicated by the red arrow on the figure) and prevents the particle to reach equilibrium in the expected time. The ESE is also very sensitive to other external perturbations, indeed a small drift in calibration may be responsible for the very small slip of the black curve under its final value at $t_f$. These observations are very reproducible and one may see in Appendix A.2 complementary results highlighting the increase of the rebound height with the intensity of the coupling.

To put it in a nutshell, Fig. 2 highlights that even though the protocol is designed to be much faster than the coupling characteristic time, the coupling perturbation impacts the response to the *basic ESE*. Our sensitive experimental setup enables us to observe experimentally the tiny effect of hydrodynamic coupling: the particle variance features a rebound at $t_f$ and will not reach equilibrium before its natural relaxation time. Nevertheless the *basic ESE* is rather robust, as for moderate coupling, this bounce is modest compared to the natural relaxation amplitude evolution. Indeed *basic ESE* still provides correct results, with a 10% deviation to

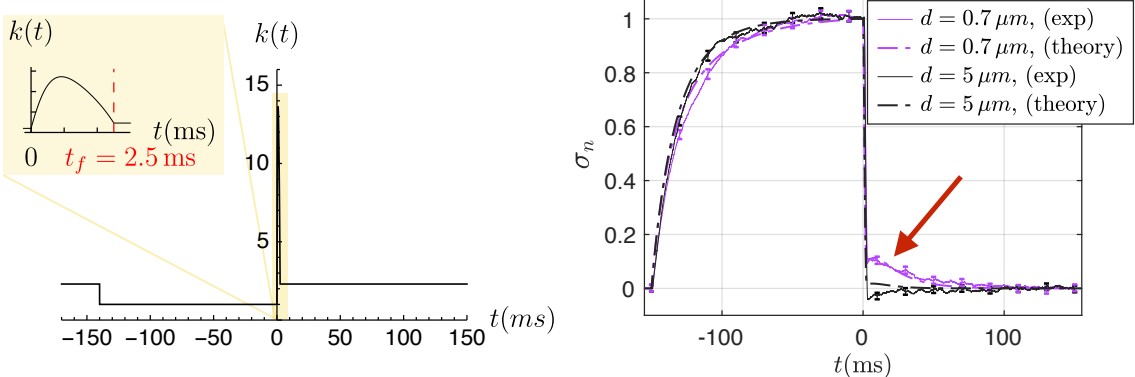

Figure 2: On the left, the stiffness profile applied to both wells: a STEP decompression at $t = -140$ ms followed by a *basic ESE* protocol for compression for $0 < t < t_f$. This procedure is called the *symmetric protocol*. At $t = -140$ ms, the stiffness jumps from $k_f = 2.3$ to $k_i = 1$. At $t = 0$ the particle is thus in its initial equilibrium when we apply an ESE protocol finishing at $t_f$ to bring back the particle to its final state at $k_f$. The ESE parameters are: $t_f = 2.5$ ms, $K_i = 4 \times 10^{-7}$ N/m, $k_f = 2.3$, and $\Gamma = 18.9$. This stiffness profile emphasizes the difference between the relaxation after a step function, and the response to the ESE protocol. On the right, normalized variance of the first particle $\sigma_n = (\sigma(t) - \sigma_f)/(\sigma_i - \sigma_f)$, corresponding to the *symmetric protocol* on the left. The plain lines are the experimental results with their error bars. The dashed curves are numerically computed from the theoretical analysis of section 4, plugging the experimental parameters from the calibration. The same process is applied for particles separated by $d = 5\,\mu$m (black) and $d = 0.7\,\mu$m (purple) corresponding respectively to a coupling constant (introduced in section 4) $\epsilon = 0.21$ and $\epsilon = 0.5$. A small rebound (around 10% of the step) pointed by the red arrow and long relaxation time are visible for close particles.

the 0-coupling case. Within this framework a measure with a poor statistics will hide the effect inside the statistical error.

It remains to be seen whether this experimental results can be supported by a theoretical analysis. To this end we devote the next section to study the coupled system's dynamics, first in equilibrium and then when driven by the *symmetric protocol*.

## 4 Theoretical analysis

To describe the evolution of two trapped brownian particles which are hydrodynamically coupled, we write the coupled Langevin equations,

$$\begin{pmatrix} \dot{x}_1 \\ \dot{x}_2 \end{pmatrix} = \mathcal{H} \begin{pmatrix} F_1 \\ F_2 \end{pmatrix}, \tag{2}$$

where $x_j$ is the position of the particle $j = 1, 2$ relative to its trapping position (see Fig. 1), $\dot{x}_j$ is the time derivative of $x_j$, and $\mathcal{H}$ is the hydrodynamic coupling tensor. The Langevin equations govern the system evolution in general whether or not it is at equilibrium. Besides, the Langevin equations (2) do not include any acceleration term: we assume the over-damped regime which is fully justified for colloidal objects (see Appendix A.3). At equilibrium the forces

acting on the particles are:

$$F_j = -K_j x_j + f_j,\tag{3}$$

where $K_j$ is the stiffness of the trap $j$ and $f_j$ is the Brownian random noise. For two identical particles of radius $r$ separated by a distance $d$ (see Fig. 1), assuming that their displacements are small compared to the mean distance between them, the hydrodynamic coupling tensor reads [7–10]:

$$\mathcal{H} = \frac{1}{\gamma} \begin{pmatrix} 1 & \epsilon \\ \epsilon & 1 \end{pmatrix}.\tag{4}$$

In some approximations described in Appendix A.4 we can write $\epsilon = \frac{3}{2}v - v^3$, where $v = r/(2r + d)$.

Let us first study how the particles behave at equilibrium ($K_j$ constant in time), and in particular how they influence their neighbour. At equilibrium the two particles are statistically independent: $\langle x_1 x_2 \rangle_{eq} = 0$, $\langle x_1^2 \rangle_{eq} = k_B T/K_1$, and $\langle x_2^2 \rangle_{eq} = k_B T/K_2$ (with $k_B$ the Boltzmann's constant and $T$ the bath temperature). However, the 2 particles are coupled by eq. (2). Extending the computation of refs. [9, 11] to the more general case of two potentials with different stiffnesses, we show in Appendix A.5 that, at equilibrium, auto-correlations $\langle x_j(0)x_j(t)\rangle$ and cross-correlations $\langle x_j(0)x_k(t)\rangle$ (with $j \neq k$) of positions read as:

$$\langle x_1(t)x_1(0)\rangle = \frac{k_B T}{2K_1 \kappa}\Big[e^{-\frac{t}{\tau_+}}(K_1 - K_2 + \kappa) + e^{-\frac{t}{\tau_-}}(K_2 - K_1 + \kappa)\Big],\tag{5}$$

$$\langle x_2(t)x_2(0)\rangle = \frac{k_B T}{2K_2 \kappa}\Big[e^{-\frac{t}{\tau_+}}(K_2 - K_1 + \kappa) + e^{-\frac{t}{\tau_-}}(K_1 - K_2 + \kappa)\Big],\tag{6}$$

$$\langle x_1(t)x_2(0)\rangle = \frac{\epsilon k_B T}{\kappa}\Big[e^{-\frac{t}{\tau_+}} - e^{-\frac{t}{\tau_-}}\Big],\tag{7}$$

with

$$\kappa = \sqrt{(K_1 - K_2)^2 + 4\epsilon^2 K_1 K_2},\tag{8}$$

$$\tau_- = \frac{2\gamma}{K_1 + K_2 - \kappa},\tag{9}$$

$$\tau_+ = \frac{2\gamma}{K_1 + K_2 + \kappa}.\tag{10}$$

We report the computed behaviour in Fig. 3. Those correlation functions involve two characteristic times $\tau_+$ and $\tau_-$ that are very close to the natural relaxation time of the harmonic well $\tau_{\text{relax}} = \gamma/K_1 \sim 15\,\text{ms}$. We consequently introduce a slow mode and a fast mode associated respectively with $\tau_-$ and $\tau_+$. The slow mode vanishes when $x_1 \propto x_2$, and the fast mode when $x_1 \propto -x_2$: ie correlation enhances the fast mode (correlated mode) and anti-correlation the slow mode (anti-correlated mode). In the symmetric case, the two modes may be interpreted as the barycentre of the system $x_M = (x_1 + x_2)/2$, and the particles separation $x_\mu = (x_2 - x_1)/2$. Naturally, $x_\mu$ embodies the slow mode and $x_M$ the fast one, as the evolution of $x_\mu$ requires a fluid displacement between the particles, while the barycentre evolution relies on the fact that one sphere tends to drag the other in its wake (details in [9]). As far as auto-correlation functions are concerned, the shape of decaying exponential in Fig. 3 is rather common. The negative cross-correlation might however be surprising. This feature stems first from the fact that the cross-correlation has to vanish at $t = 0$ (a consequence of independence at equilibrium), and second from the fact that the anti-correlated mode (associated with $x_\mu$) lives longer than the correlated mode (associated with $x_M$).

We now focus on the dynamics of the particles when the potentials change with time. It proves convenient to convert the coupled Langevin equations into equations describing the

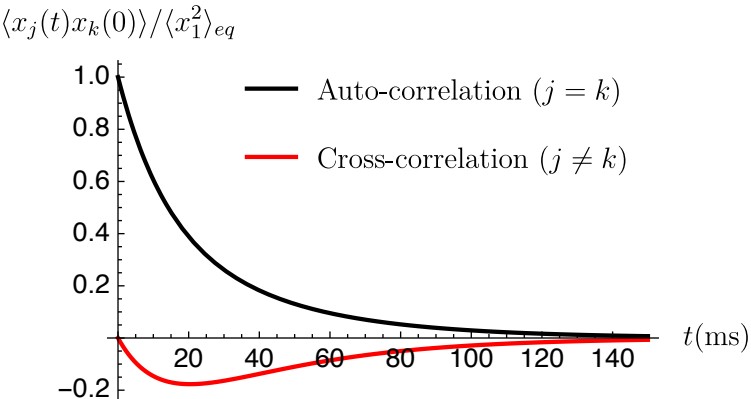

Figure 3: Auto- and cross-correlation functions normalized by $\langle x_1^2 \rangle_{eq}$ as a function of time, when $\gamma = 1.88 \times 10^{-8}$ sN/m, $K_1 = K_2 = 10^{-6}$ N/m and $d = 1\,\mu$m, so that $\sigma_{11,eq} = k_B T / K_1 = 4 \times 10^3$ nm$^2$ and $\epsilon = 0.46$. We recover at $t = 0$ the values of the moments at equilibrium, in particular $\sigma_{12,eq} = 0$.

dynamics of the moments $\langle x_1^2 \rangle(t)$, $\langle x_2^2 \rangle(t)$ and $\langle x_1 x_2 \rangle(t)$. Using the dimensionless quantities $\sigma_{jk} = K_{1,i} \langle x_j x_k \rangle / (2k_B T)$, we obtain the following system to describe the evolution of the moments (see Appendix A.6):

$$\Gamma \frac{d\sigma_{11}}{ds} = -2k_1 \sigma_{11} - 2\epsilon k_2 \sigma_{12} + 1, \tag{11}$$

$$\Gamma \frac{d\sigma_{22}}{ds} = -2k_2 \sigma_{22} - 2\epsilon k_1 \sigma_{12} + 1, \tag{12}$$

$$\Gamma \frac{d\sigma_{12}}{ds} = -(k_1 + k_2)\sigma_{12} - \epsilon(k_2 \sigma_{22} + k_1 \sigma_{11} - 1), \tag{13}$$

where $s = t/t_f$ as before, $k_j(s) = K_j(s)/K_{1,i}$ ($K_{1,i}$ being the initial stiffness of the first well), and $\Gamma = \gamma/(K_{1,i} t_f)$. The above equations contain all the information about the dynamics of the system, as the joint probability distribution remains Gaussian out of equilibrium (see Appendix A.7) and is thus fully described by $\sigma_{11}$, $\sigma_{22}$ and $\sigma_{12}$. The *basic ESE* in eq. (1) is defined in ref. [6] using eq. (11) without the cross term $\epsilon\sigma_{12}$ term. Therefore it cannot be operational for the coupled system.

We compute numerically the evolution of the first particle variance corresponding to the *symmetric protocol* where the stiffness of both wells is simultaneously driven according to the *basic ESE*. The results of these computations are summarized in Fig. 4: it should be recalled that in the *symmetric protocol* context ($K_1 = K_2 = K$), the above equations simplify and $\sigma_{11} = \sigma_{22}$ can be written $\sigma$.

The theoretical predictions of Fig. 4 seem to be consistent with the experimental conclusions drawn in section 3. To confirm that the model prediction and the experimental curves match, we superimpose in dashed lines on Fig. 2 the theoretical curves obtained using the same ESE parameters and the external parameters from calibration. We see that the results are in very good accordance. Besides, the validity of the theory during the STEP to prepare the system at $K_i$ confirms that the calibration is relevant to estimate the external parameters during the experiment.

The model of the hydrodynamic coupling proves to be precise enough to be used for ESE computations. We are thus equipped to propose a new strategy to drive a coupled system without any compromise on the shortcut efficiency. Indeed we can take into account the hydrodynamic coupling in the construction of a new ESE protocol thereby eliminating the small although spurious bounce identified above.

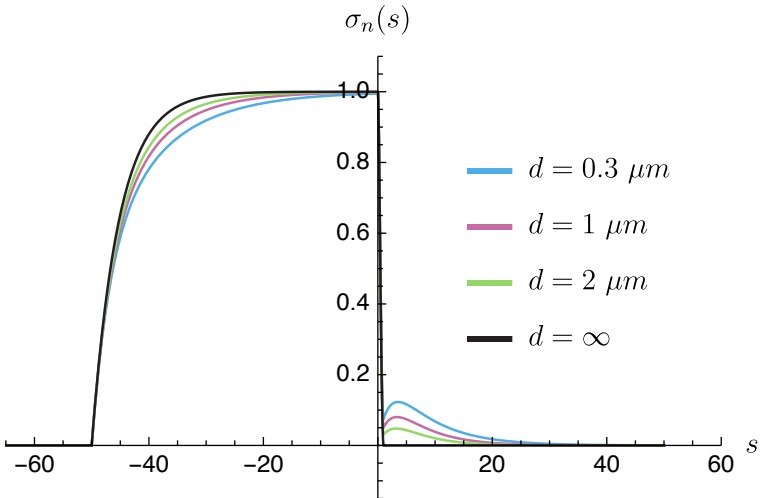

Figure 4: Evolution of the normalized variance $\sigma_n = (\sigma(t) - \sigma_f)/(\sigma_i - \sigma_f)$ of one particle in response to the *symmetric protocol* for different values of the distance $d$ between the particles. The parameters of the ESE protocol (shaped as in Fig. 2) are the following: $t_f = 2\,\text{ms}$, $k_f = K_f/K_i = 1.5$, $K_i = 10^{-6}\,\text{N/m}$, and $\Gamma = 9.42$. Without coupling (when $d = \infty$) the response to the ESE is shortcut to $t_f$. The hydrodynamic coupling results in a rebound on the variance curve, which no longer reaches its equilibrium value at $t_f$, but after a few natural relaxation times $\tau_{\text{relax}} \approx 15\,\text{ms}$. As expected from experimental results, the smaller the distance $d$, the higher the rebound and so the deviation from the 0-coupling case.

## 5   Coupled ESE protocol

Our strategy to design a *coupled protocol* is now to look for an ESE scheme that would drive the first particle from $(t_i = 0, K_i)$ to $(t_f, K_f)$ while being robust to coupling interaction. A solution to achieve this requirement is to design a protocol that does not depend on the coupling intensity (ie independent of the $\epsilon$ parameter). This strong constraint can be met if we require particle independence at all time, that is to say $\langle x_1 x_2 \rangle(t) = 0$ during all the process and not only at equilibrium states. Indeed insofar as we require independence, the results no longer depend on the strength of the coupling.

   As detailed in Appendix A.8, the independence requirement ($\sigma_{12} = 0$ during the process) enables us to simplify the evolution equations eq. (11)-(13) and to find an ESE protocol that meets the requirements detailed above: we find a shape for $k_1(s)$ and $k_2(s)$ independent of $\epsilon$ that satisfies the equilibrium at $t_f$ of both particles (see Fig. 5). The expression of $k_1(s)$ is therefore the same as in the single particle case, but the second potential has to be driven appropriately with a different stiffness profile $k_2(s)$.

   The price to pay to drive the particle 1 from $K_{1,i}$ to $K_{1,f}$ is to enforce a nearly opposite profile on the second potential. In particular the final value of the second well stiffness $K_{2,f}$ is imposed by the parameters chosen for the first particle and is therefore not chosen a priori. Besides, a sum rule ensues, such that $k_1 \sigma_{11} + k_2 \sigma_{22}$ is conserved. To maintain independence, the two wells tend to evolve in opposition because of the correlation due to the coupling. Indeed the coupling term $\epsilon F_1$ in eq. (2) can be interpreted as an extra random noise:

$$\gamma \dot{x}_2 = -K_2 x_2 + f_2 + \epsilon F_1. \tag{14}$$

This coupling term behaves as the random noises with the following characteristics (at equi-

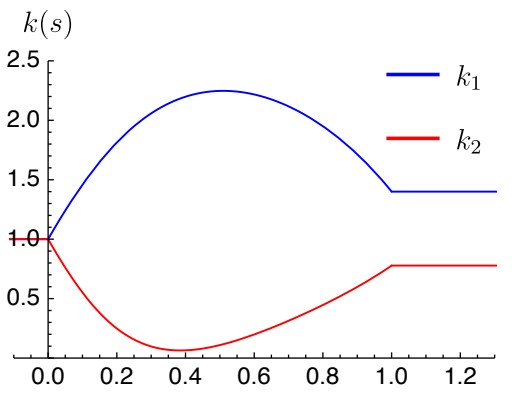 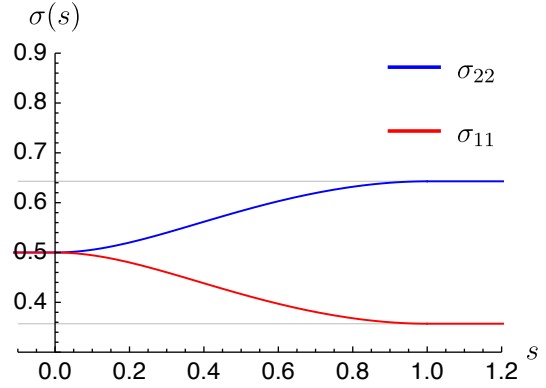

Figure 5: (Left) Profiles $k_1(s)$ and $k_2(s)$ computed for the *coupled ESE protocol* that maintains independence between the particles for parameters: $k_f = 1.4$, $K_i = K_{2,i} = 1.8 \times 10^{-6}$ N/m, $t_f = 2.5$ ms, and $\Gamma = 4.19$. While $k_1$ (red) reaches the target value at $s = 1$, the second well stiffness $k_2$ (blue) has to adapt itself. In particular its final stiffness value is determined by the other parameters of the ESE: $k_{2f} = k_{2i}k_f/(k_{2i}k_f + k_f - k_{2i})$. This protocol does not depend on the coupling constant $\epsilon$ and so works for any distance $d$ between the particles. (Right) Result of the computation for the dimensionless variances of the two particles using the ESE protocol on top: $\sigma_{11}$ in red, and $\sigma_{22}$ in blue. The plot confirms that Boltzmann equilibrium (horizontal grey lines) is reached for both particles at initial and final times. Let us remind that $\sigma_{12} = 0$ all along.

librium),

$$\langle \epsilon F_1 \rangle = -\epsilon K_1 \langle x_1 \rangle + \epsilon \langle f_1 \rangle = 0, \tag{15}$$

$$\langle \epsilon^2 F_1^2 \rangle = \epsilon^2 k_1^2 \langle x_1^2 \rangle + \epsilon^2 \langle f_1^2 \rangle = \epsilon^2 k_B T k_1 + \epsilon^2 \langle f_1^2 \rangle. \tag{16}$$

Thus if $k_1$ increases, the noise imposed to particle 2 by the coupling increases as well, and consequently so does the variance of particle 2. To pretend that the two particles are independent and that this increase in the particle 2 variance is not due to the behaviour of the particle 1, the second well should open up. That is why to maintain a vanishing cross term $\sigma_{12} = 0$ the second well should behave in opposition to the first one (see Fig. 5).

The experimental implementation of the *coupled protocol* is illustrated in Fig. 6. The distance between the particles is set to $d = 0.8\,\mu$m to ensure strong coupling. We compare the response of the system to the *symmetric protocol* in which the two potentials are driven similarly, with the response to the *coupled ESE*.

In this new set of experiments, the rebound in response to the *symmetric protocol* is naturally still present, but disappears when applying the *coupled ESE protocol*. This result validates the efficiency of enforcing independence for coupled particles. Indeed this protocol is very stable against the coupling interaction because it does not depend on the strength of the coupling ($\epsilon$ in our model). Thanks to this process we achieve the same efficiency of shortcut to equilibrium we had for a single particle, but now for coupled ones. This extension of the validity of ESE protocol has nevertheless a cost: the second particle, coupled to the particle of interest, has to be driven to a final equilibrium state defined by the other parameters of the protocol ($K_{2i}$ and $k_f$).

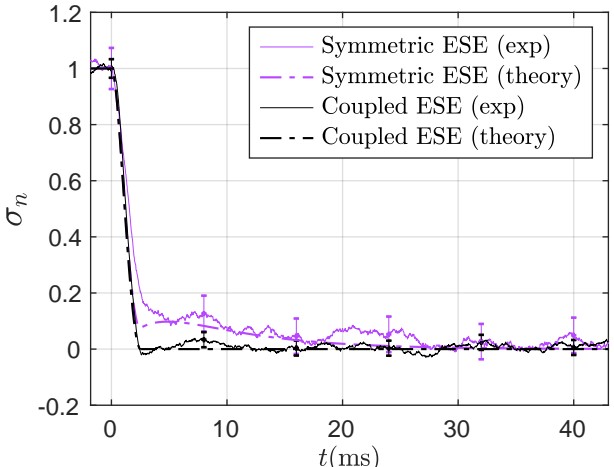

Figure 6: Normalized variance $\sigma_n = (\sigma(t) - \sigma_f)/(\sigma_i - \sigma_f)$ of the first particle when the potentials are driven by the *symmetric protocol* (purple) or by the *coupled ESE protocol* (black). The parameters of the experiment are: $k_f = 1.4$, $K_{1,i} = K_{2,i} = 1.8 \times 10^{-6}\,\text{N/m}$, $t_f = 2.5\,\text{ms}$, $d = 0.8\,\mu\text{m}$, and thus $\epsilon = 0.49$ and $\Gamma = 4.19$. The *symmetric protocol* leads to the rebound predicted in section 4. On the contrary the *coupled protocol* designed to cancel the correlations between the particles works as expected: the rebound is essentially suppressed and the particle reaches equilibrium at $t_f$. Furthermore, the experimental results (plain lines) are again consistent with the theoretical predictions (dashed lines) based on measured parameters only and not on adjustable ones.

## 6 Limits and other approaches

We are experimentally facing two limitations in the implementation of the *Coupled ESE*. First, stiffnesses have to remain positive (ie attractive potentials), and second they cannot exceed maximum values above which the particles can be damaged. Actually it is possible to mimic repulsive potentials and go beyond the first constraint [12], but considering our basic optical tweezers set up, it is far more convenient to stick to positive stiffness. In the case of the *Coupled ESE*, assuming that $k_{2,i} = 1$ and $k_f > 1$, these limitations translate into $k_2 > 0$ and $k_1 < k_{max}$. Using the expression of $k_2(s)$ and $k_1(s)$ the first limit can be expressed as a constraint on the acceleration factor $\Gamma$, or equivalently on $t_f$ and $K_i$ as $\Gamma = \gamma/(K_i t_f)$. Indeed maintaining $k_2 > 0$ requires

$$\Gamma < \Gamma_{\text{lim},1} = \min[-\frac{1}{\dot{\sigma}_{11}(s)}, 0 < s < 1].\tag{17}$$

$\Gamma_{\text{lim},1}$ depends on $k_f$ (yellow curve in Fig. 7): the more one wants to compress the well, the smaller $\Gamma$ should be, and so the higher $t_f$ will be.

Concerning the second limit $k_1 < k_{max}$ a similar computation gives us the corresponding constraint on $\Gamma$. We introduce:

$$\Gamma_{\text{lim}}(s) = \frac{((k_f - 1)s^2(2s - 3) - 1)(k_{max} - 1 + (k_f - 1)s^2(2s - 3))}{3(k_f - 1)s(s - 1)},\tag{18}$$

Then,

$$\Gamma_{\text{lim},2} = \min[\Gamma_{\text{lim}}(s), 0 < s < 1].\tag{19}$$

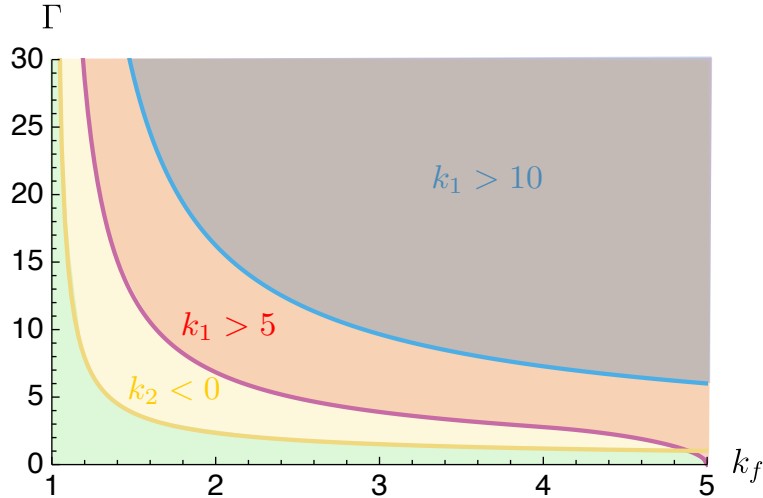

Figure 7: Experimental limits of the *coupled ESE protocol* in terms of the speed-up parameter $\Gamma$ for a compression of the first particle ($k_f > 1$). The yellow line represents the higher limit $\Gamma$ should not exceed to maintain $k_2 > 0$, the red one to maintain $k_1 < 5$ and the blue one to maintain $k_1 < 10$. The requirement $k_2 > 0$ being the most restrictive, the limit to respect during experiments is the yellow line that corresponds to $\Gamma_{lim,1}$. In other words, the working region where all the constraints are met is the green region. The yellow, red and blue regions delineate the domains where the respective requirements are not met anymore.

To summarize, we plot in the Fig. 7 the maximum boundary $\Gamma_{\mathrm{lim}}$ to comply with the constraints $k_2 > 0$ (yellow curve) and $k_1 < k_{\mathrm{max}}$ for $k_{\mathrm{max}} = 5$ (red curve) and $k_{\mathrm{max}} = 10$ (blue curve). As expected, the stronger is the compression, the smaller is the region accessible for $\Gamma$, because it has to remain under $\Gamma_{\mathrm{lim}}$. The limit $k_2 > 0$ is the most restrictive, and that is why $\Gamma_{\mathrm{lim},1}$ in yellow delimits the working region. To provide shortcuts outside the accessible region, some new strategies should be developed such as what has been done in ref. [13] for the *basic ESE*.

Enforcing independence through the *coupled ESE protocol* is a successful strategy to extend the family of ESE protocols to more complex systems which cannot be managed with full efficiency by the *basic ESE*. Within the limits we highlighted above, this particular solution independent of $\epsilon$ turns out to be very powerful. Yet, the solution panel to the coupled case problem is wide, and there is more to find in this direction. In particular, it is possible to guide the two particles with the same stiffness profile to a chosen target state. This *symmetric coupled ESE protocol* detailed in Appendix A.9 has nevertheless a cost: cross-correlations appear during the process and vanish only at equilibrium. Therefore, the independency is no longer required in this protocol, which makes it depend on the coupling intensity. That is why this $\epsilon$ dependent protocol is harder to implement experimentally. Further work is required to extend ESE protocols to more complex systems, and every solution will have specific advantages and limits.

# 7 Conclusion

In conclusion, we explored shortcut to adiabadicity schemes for coupled systems: in particular two hydro-dynamically coupled particles. The first objective of this paper was to test the

stability of the *basic ESE protocol* designed for single systems against the coupling interaction. Our experiments, in very good accordance with the model, proved its relative robustness: the coupling perturbation deviates the response of a dozen of percents compared to the 0-coupling case. It is nevertheless possible to work out explicitly ESE solutions that take due account of the coupling, and are therefore immune to it: this is the second message of this article. The model used to describe the coupling proved reliable enough to build a new family of ESE solutions with the same method of retro-computing used to find the single particle ESE protocol. We thus propose a very robust protocol, because $\epsilon$ independent, that enforces independence between the particles. Experimental tests confirm the efficiency of this shortcut strategy within the experimental limits described in the last part of the paper. Other solutions can be investigated such as a symmetric protocol designed for coupled particles (more difficult to implement because $\epsilon$ dependent).

# Acknowledgements

We thank Loïc Rondin for interesting discussions.

**Funding information** This work has been financially supported by the Agence Nationale de la Recherche through grant ANR-18-CE30-0013.

**Supporting material** Experimental datasets and codes are publicly available on doi:10.5281/zenodo.4242922.

# A Appendix

## A.1 Calibration procedure

As the effect under scrutiny is tiny, a very accurate calibration is necessary to observe it. Thus we detail in this section the calibration procedure conducted before the experimental tests of ESE protocols. It is performed as follows: first we have to find the connection between the amplitude $A$ of the sine wave driving the AOD and the stiffness $K$ applied by the optical trap to the particle. To do so, we acquire the position variance ($\sigma^2 = k_B T/K$) for different amplitudes $A$. This calibration curve enables us to convert the ESE protocol in driving amplitude for the AOD. Then, the only dependence on external parameters of the ESE protocol lies in the parameter $\Gamma = \gamma/(K_i t_f)$. To estimate $\Gamma$ we conduct the acquisition of the cut off frequency [14] ($f_0 = K/(2\pi\gamma)$) when the particle is in the initial state of the ESE, $f_{0,i}$, through the particle's Brownian noise spectrum in position corresponding to the initial value of amplitude $A_i$. Then we deduce $\Gamma = 1/(2\pi f_{0,i} t_f)$.

One may now wonder to what extent small drifts in calibration may impact the experimental results. Indeed during the typical time of our experiments (up to a few hours), we observe that the stiffness $K$ and the parameter $\Gamma$ decrease by a small amount: 4% at most. The stiffness variation can be a consequence of the variation of the AOD efficiency because the AOD warms up with time. On the other side, $\Gamma$ is modified because of the following phenomena: the stiffness variation, the water viscosity dependency on the temperature, and the damping coefficient correction due to the distance $h$ to the cell walls. Indeed at first order in $r/h$ we can expand [15] $\gamma(T, h) = 6\pi r \eta_0(T) \times (1 + 9/16 \times r/h)$, with $\eta_0(T)$ decreasing of 2% per Kelvin, and the term in $r/h$ leading to an additional 1% per 5 $\mu$m in $h$.

Those variation in $K$ and $\Gamma$ are small, leading to a small error on the ESE protocols themselves. Moreover, our cycle procedure of acquisition makes the comparison of protocols in equivalent experimental conditions. Drifts in $\Gamma$ have the same consequences on the different responses we compare: the relative differences between the curves are only weakly sensitive to variations in $\Gamma$. Finally, drifts in $K_i$, $K_f$ (thus $\sigma_i$, $\sigma_f$) are wiped out by plotting the normalised variance.

Furthermore, the local drift of the bath temperature due to the power of the lasers (measuring laser and trapping laser), amplifies the deviation of the particle variance also affected by the stiffness drift. Indeed the standard deviation $\sigma$ can increase up to 2% during an acquisition. As we are studying $\sigma$ jumps of 20% with ESE, it is better to get rid of the 2% error due to external parameters small deviations. To do so, we normalize the results at regular time intervals to minimize the drift effect in the results.

## A.2 Complementary experimental results

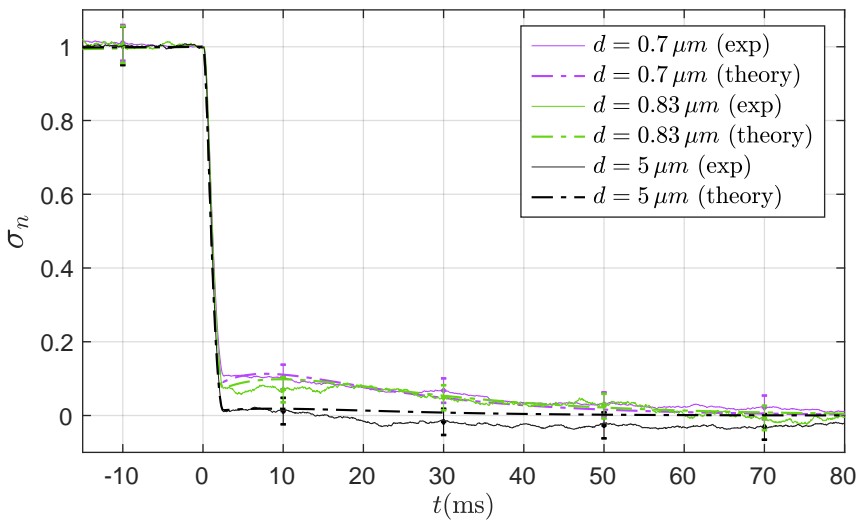

Figure 8: Same as Fig. 2 but with three distances: $d = 5\,\mu$m in black, $d = 0.83\,\mu$m in green and $d = 0.7\,\mu$m in purple.

As a complement to the results presented in Fig. 2, we propose another experimental result in Fig. 8. All the parameters are the same as in Fig. 2 but the experiment is performed with 3 different distances between the particles. From it, we can affirm first that the results are very reproducible and always consistent with the theory, and second that the rebound decreases with the coupling as pointed out in Fig. 4.

## A.3 Over-damped regime

The influence of the inertia lasts on a characteristic time $\tau_{\text{inertia}} = m/\gamma = 2\mu r^2/(9\eta)$, with $\mu$ being the volumic mass of the particles. As we consider usual fluids such as water, $\eta = 10^{-3}\,\text{Pa s}$, and $\mu = 10^3\,\text{kg m}^{-3}$. The point is then to compare $\tau_{\text{inertia}}$ with the time needed for the particle to diffuse over a distance equivalent to its diameter, $\tau_{\text{diff}}$. In a usual diffusion process we have, $\tau_{\text{diff}} = (2r)^2/D$, using the diffusion coefficient $D = k_B T/(6\pi\eta r)$. Therefore, on the one hand, the $r$ region where $\tau_{\text{inertia}} \ll \tau_{\text{diff}}$ corresponds to $r \gg 0.01\,\text{pm}$.

On the other hand, to get an upper limit, we compare $\tau_{\text{inertia}}$ to the characteristic time of the experiment $\tau_{ESE} = 1\,\text{ms}$. Indeed in the context of shortcuts, the time of the ESE is more restric-

tive than the natural relaxation time $\tau_{\text{relax}} = \gamma/K \sim 15\,\text{ms}$. The assumption $\tau_{\text{inertia}} \ll \tau_{ESE}$ remains valid while $r \ll 70\,\mu\text{m}$. To conclude, the $r$ region of the over-damped regime is $0.01\,\text{pm} \ll r \ll 70\,\mu\text{m}$.

We are thus working in the $r$ region where the inertia faded too fast as compared to the other phenomena to be noticed (indeed for $r = 1\,\mu\text{m}$, $\tau_{\text{inertia}} \sim 0.2\,\mu\text{s}$ and $\tau_{\text{diff}} \sim 20\,\text{s}$): the regime is over-damped.

## A.4 Model for Hydrodynamic coupling

The hydrodynamic interactions of the particles with the surrounding fluid are described by their mobility matrix $\mathcal{H}$ (eq. (4)), which is also known as the Rotne-Prager diffusion tensor [7–9]. The Rotne-Prager diffusion tensor consists in adding third order correction in $(r/d)^3$ to the off-diagonal elements of the Oseen tensor. Under our experimental conditions, this corrections is always smaller than 3.5%. The form of the coupling parameter $\epsilon$ depends on different approximations. Here we assume $\epsilon$ to be constant: it involves only the distance between the wells $d$ and not the distance between the particles $(x_1 - x_2)(t)$. This assumption is supported by the following order of magnitudes: one particle can diffuse up to its rms displacement $\delta x_{\text{rms}} = \sqrt{k_B T/k} \sim 60\,\text{nm} \ll d$, so that in first approximation $|x_1 - x_2| = d$ and $\epsilon = f(d)$. The expression of $\epsilon = f(d)$ is given by the Rotne-Prager approximation: for particle distances larger than $d = r$, we can write $\epsilon = \frac{3}{2}\nu - \nu^3$, where $\nu = \frac{r}{d}$. The term $\nu$ becomes more important when particles are close to each other. At very short distances, when $d \lesssim r/10$, lubrication forces would have to be taken into account explicitly. On the contrary, in the small $\nu$ limit, we reach the Oseen approximation where $\epsilon = \frac{3}{2}\nu$.

## A.5 Auto and Cross-Correlation

We start from the coupled Langevin equations (2):

$$\gamma \dot{x}_1 = -K_1 x_1 - \epsilon K_2 x_2 + f_1 + \epsilon f_2 , \tag{20}$$
$$\gamma \dot{x}_2 = -K_2 x_2 - \epsilon K_1 x_1 + f_2 + \epsilon f_1 , \tag{21}$$

and we use the Laplace Transform:

$$\widehat{x}(s) = \int_0^{+\infty} x(t)e^{-st} dt . \tag{22}$$

After having Laplace transformed the system (20), (21) we obtain (to simplify we stop indicating variables $s$ and $t$, $\widehat{x}$ transformed functions implies $s$ variable, and $x$ functions $t$):

$$\gamma(s\widehat{x_1} - x_1(0)) = -k_1 \widehat{x_1} - \epsilon k_2 \widehat{x_2} + \widehat{f_1} + \epsilon \widehat{f_2} , \tag{23}$$
$$\gamma(s\widehat{x_2} - x_2(0)) = -k_2 \widehat{x_2} - \epsilon k_1 \widehat{x_1} + \widehat{f_2} + \epsilon \widehat{f_1} . \tag{24}$$

We then multiply the two above equations by $x_2(0)$ and take the mean value:

$$\gamma(s\langle \widehat{x_1} x_2(0)\rangle - \sigma_{12}^2) = -k_1 \langle \widehat{x_1} x_2(0)\rangle - \epsilon k_2 \langle \widehat{x_2} x_2(0)\rangle ,$$
$$\gamma(s\langle \widehat{x_2} x_2(0)\rangle - \sigma_{22}^2) = -k_2 \langle \widehat{x_2} x_2(0)\rangle - \epsilon k_1 \langle \widehat{x_1} x_2(0)\rangle .$$

This system is now easy to solve (knowing the values of $\sigma_{22}$ and $\sigma_{12}$ at equilibrium at $t = 0$). The last step only consists in taking the Inverse Laplace Transform of the expressions obtained, that leads to the expression of $\langle x_1(t)x_2(0)\rangle$ and $\langle x_2(t)x_2(0)\rangle$ of eqs. (7) and (6). We can reproduce the procedure by multiplying this time by $x_1(0)$ to obtain the expression of $\langle x_1(t)x_1(0)\rangle$ of eq. (5).

## A.6  Evolution of the moments

To meet the Boltzmann equilibrium prediction the random noises $f_j$ in eq. (2) and in eqs. (20)-(21) should verify:

$$\langle f_1(0)f_1(t)\rangle = 2k_B T\gamma\frac{1}{1-\epsilon^2}\delta(t) = \langle f_2(0)f_2(t)\rangle\,, \tag{25}$$

$$\langle f_1(0)f_2(t)\rangle = -2k_B T\gamma\frac{\epsilon}{1-\epsilon^2}\delta(t)\,. \tag{26}$$

Then, starting with the coupled Langevin equation (2), we want to deduce the evolution of the moments of the joint probability in position. To do so we follow the Ito prescription ($\langle f_1(t)x_1(t)\rangle = 0$) and apply the Ito chain rule on $x_1^2(t)$. Combined with equation (2), and after taking the mean value, we obtain:

$$\gamma\langle x_1\frac{dx_1}{dt}\rangle = -K_1\langle x_1^2\rangle - \epsilon K_2\langle x_1 x_2\rangle + \epsilon^2\langle f_2^2\rangle + \langle f_1^2\rangle + 2\epsilon\langle f_1 f_2\rangle\,. \tag{27}$$

Using the auto-correlation values of the $f_j$'s in (25) and (26), we readily obtain:

$$\frac{\gamma}{2}\frac{d\langle x_1^2\rangle}{dt} = -K_1\langle x_1^2\rangle - \epsilon K_2\langle x_1 x_2\rangle + kT\,. \tag{28}$$

Finally we reproduce the procedure for the other moments and using again dimensionless quantities ($\sigma_{jk} = \langle x_j x_k\rangle\frac{K_{1,i}}{2k_B T}$) we obtain the system to describe the dynamics of the moments given above in eqs. (11), (12) and (13).

## A.7  Gaussian behaviour of the coupled particles joint probability distribution

Similarly to the single particle case, we can describe the system through the evolution of its probability density to find the first particle in $x_1$ and the second in $x_2$ at time $t$, $P(x_1, x_2, t)$. The time evolution of the joint Probability $P(x_1, x_2, t)$ is governed by the Fokker-Planck equation:

$$\frac{\partial P}{\partial t} = -\sum_{j=1}^{j=2}\frac{\partial g_j P}{\partial x_j} - \sum_{j,k=1}^{j,k=2}\theta_{jk}\frac{\partial^2 P}{\partial x_j \partial x_k}\,, \tag{29}$$

where,

$$g_1 = -\frac{1}{\gamma}K_1 x_1 - \frac{\epsilon}{\gamma}K_2 x_2\,, \tag{30}$$

$$g_2 = -\frac{1}{\gamma}K_2 x_2 - \frac{\epsilon}{\gamma}K_1 x_1\,, \tag{31}$$

$$\theta_{jj} = \frac{k_B T}{\gamma}\,, \tag{32}$$

$$\theta_{jk} = \frac{k_B T\epsilon}{\gamma}\ \text{for}\ j \neq k\,. \tag{33}$$

In order to prove the Gaussian behaviour of the joint Probability, we propose a 2D generalisation of the computation made in ref. [16]. We introduce the 2D Fourier Transform:

$$G(p_1, p_2, t) = \iint_{-\infty}^{+\infty} e^{ip_1 x_1}e^{ip_2 x_2}P(x_1, x_2, t)dx_1 dx_2\,. \tag{34}$$

We apply this Fourier Transform to Fokker-Plank eq. (29)

$$\frac{\partial G}{\partial t} = -\frac{K_1(p_1 + \epsilon p_2)}{\gamma}\frac{\partial G}{\partial p_1} - \frac{K_2(p_2 + \epsilon p_1)}{\gamma}\frac{\partial G}{\partial p_2} - \frac{k_B T}{\gamma}G[(p_1^2 + p_2^2) + \epsilon p_1 p_2], \tag{35}$$

$$\frac{\partial \ln G}{\partial t} = -\frac{K_1(p_1 + \epsilon p_2)}{\gamma}\frac{\partial \ln G}{\partial p_1} - \frac{K_2(p_2 + \epsilon p_1)}{\gamma}\frac{\partial \ln G}{\partial p_2} - \frac{k_B T}{\gamma}[(p_1^2 + p_2^2) - 2\epsilon p_1 p_2]. \tag{36}$$

On the one hand, the expansion of $G$ generates the moments $\mu_{n,m} = \langle x_1^n x_2^m \rangle$, since $G(p_1, p_2, t) = \sum_{n,m=0}^{+\infty}(ip_1)^n(ip_2)^m \mu_{n,m}(t)/n!m!$. On the other hand the expansion of $\ln(G)$ generates the cumulants $\chi_{n,m}(t)$:

$$\ln G(p_1, p_2, t) = \sum_{n,m=1}^{+\infty}\frac{(ip_1)^n(ip_2)^m}{n!m!}\chi_{n,m}(t). \tag{37}$$

In particular, the two first cumulants in $n$ are the mean and the variance of the first particle position: $\chi_{1,0} = \mu_{1,0} = \langle x_1 \rangle = 0$ and $\chi_{2,0} = \mu_{2,0} - \mu_{1,0}^2 = \langle x_1^2 \rangle - \langle x_1 \rangle^2 = \langle x_1^2 \rangle$. Thus we identify the power of $p_1$ and $p_2$ in eq. (36) and we deduce:

$$\gamma\dot{\chi}_{nm} = -(nK_1 + mK_2)\chi_{nm} - \epsilon(mK_1\chi_{n+1,m-1} + nK_2\chi_{n-1,m+1})$$
$$+ 2k_B T(\delta_{n,2}\delta_{m,0} + \delta_{m,2}\delta_{n,0} + \epsilon\delta_{m,1}\delta_{n,1}). \tag{38}$$

For $(n,m) = (2,0)$ (that corresponds to $\sigma_{11}$), $(n,m) = (0,2)$ ($\sigma_{22}$), and $(n,m) = (1,1)$ ($\sigma_{12}$), we recover the evolution equations eq. (11)-(13). But in addition, eq. (38) for $(n+m) > 2$ entails that an initially Gaussian distribution remains Gaussian at all times. Indeed it can be easily deduced that if $\chi_{n,m}(0) = 0$ for all $(n+m) > 2$ in the equilibrium state, we have $\chi_{n,m}(t) = 0$ for all time for all $(n+m) > 2$.

## A.8  Coupled ESE enforcing independence

Requiring particle independence at all times consists in demanding $\sigma_{12} = 0$. The evolution eqs. (11)-(13) can then be simplified into:

$$\Gamma\frac{d\sigma_{11}}{ds} = -2k_1\sigma_{11} + 1, \tag{39}$$

$$\Gamma\frac{d\sigma_{22}}{ds} = -2k_2\sigma_{22} + 1, \tag{40}$$

$$1 = k_2\sigma_{22} + k_1\sigma_{11}. \tag{41}$$

We straightforwardly deduce how the second particle variance is linked to the first and how the two stiffness profiles are related,

$$\sigma_{22}(s) = -\sigma_{11}(s) + \frac{1}{2} + \frac{1}{k_{2i}}, \tag{42}$$

$$k_2(s) = \frac{2k_{2i}(1 - k_1(s)\sigma_{11}(s))}{k_{2i} - 2k_{2i}\sigma_{11}(s) + 2}. \tag{43}$$

Moreover, we observe that eq. (39) that describes the $\sigma_{11}$ evolution is the same as in the single particle case. Thus if the same ESE profile is imposed on $k_1(s)$, the equilibrium requirements on the 1st particle will be met. The corresponding $k_2(s)$ can be deduced from eq. (43). We finally obtain for the coupled particles ESE protocol:

$$k_1(s) = 1 + (k_{1f} - 1)(3 - 2s)s^2 - \frac{3\Gamma(k_{1f} - 1)(s-1)s}{1 + (k_{1f} - 1)(3 - 2s)s^2}, \tag{44}$$

$$k_2(s) = 1 + (k_{1f} - 1)(3 - 2s)s^2$$
$$+ \frac{3\Gamma(k_{1f} - 1)(s-1)s}{1 + (k_{1f} - 1)(3 - 2s)s^2}\frac{k_{2i}}{1 + (1 + k_{2i})(k_{1f} - 1)(3 - 2s)s^2}. \tag{45}$$

### A.9 Symmetric coupled ESE solution

We explored a new family of ESE solutions adapted to the coupled system by proposing the *coupled ESE* that enforces independence between the particles. But it was at the expense of having the evolution of particle 2 enslaved to that of particle 1, and thereby not a priori controlled. This results in the fact that the two particles cannot be treated symmetrically. It is thus interesting to look for another solution to the coupled problem: an ESE protocol that jointly drives the two potentials and treats the two particles in a symmetric fashion. Contrary to the *coupled ESE*, such a protocol will introduce cross-correlations between particles.

Now that we require for all time $K_1(t) = K_2(t) = K(t)$ (and so $\sigma_{11}(t) = \sigma_{22}(t)$), two modes arise from evolution equations, $u = \sigma_{11} + \sigma_{12}$ and $v = \sigma_{11} - \sigma_{12}$ that satisfy the following decoupled system:

$$\Gamma \frac{du}{ds} = -2k(s)(1+\epsilon)u(s) + (1+\epsilon), \tag{46}$$

$$\Gamma \frac{dv}{ds} = -2k(s)(1-\epsilon)v(s) + (1-\epsilon). \tag{47}$$

The modes evolve following the same form of equation with 2 different time scales $\tau_u < \tau_v$ that correspond to the $\tau_-$ and $\tau_+$ appearing into the correlation functions for the symmetric case. Indeed one may notice that $u = \sigma_{11} + \sigma_{12} = 2\langle x_M^2 \rangle$ and $v = \sigma_{11} - \sigma_{12} = 2\langle x_\mu^2 \rangle$. We naturally recover the modes corresponding to the barycentre and the particles separation evolution, with the barycentre moving faster because it does not require displacement of the fluid between the particles to do so.

The strategy to outline an ESE protocol from eqs. (46)-(47) is the following: first we propose a fifth order polynomial form of $v(s)$ with one degree of freedom (called parameter $p$) satisfying initial and final conditions of equilibrium. Secondly, we find the expression of $u(s)$ as a function of $v(s, p)$:

$$u(s) = \frac{1}{I(s)} \left(1 + \frac{2(1+\epsilon)}{\Gamma}\right) \int_0^s I(y)dy, \tag{48}$$

with

$$I(y) = \exp\left\{\frac{2(1+\epsilon)}{\Gamma} \int_0^y k(x)dx\right\} = \exp\left\{\frac{1+\epsilon}{1-\epsilon} \int_0^y \frac{(1-\dot{v}(x))}{v(x)}dx\right\}. \tag{49}$$

Finally, we tune the parameter $p$ of the ansatz of $v(s)$ to satisfy boundary conditions for $u(s)$ from eq. (48). A simple procedure of dichotomy that iterates on the value of the $p$ parameter does the job. Knowing the expression of $u(s)$ and $v(s)$, the stiffness profile can be easily deduced from eq. (46).

Fig. 9 plots an example of *symmetric coupled ESE protocol* obtained with this procedure. It is important to point out that this protocol which guides jointly the two particles of a coupled system depends on the coupling intensity ($\epsilon$). This property makes it hard to implement experimentally.

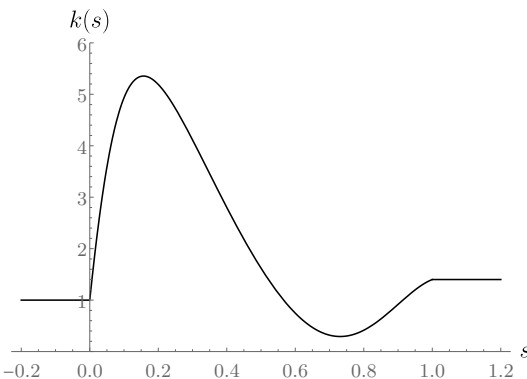

Figure 9: Stiffness profile for the symmetric coupled ESE treating particles distant by $d = 0.7\,\mu$m (coupling constant $\epsilon = 0.5$). Both potentials are controlled by the same protocol which is meant to drive the particles from $K_i$ to $K_f = k_f \times K_i$ in the desired time $t_f$. The parameters of the ESE plotted here are: $t_f = 3$ ms, $K_i = 2.5 \times 10^{-6}$ N/m, $k_f = 1.4$ and $\Gamma = 2.5$

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
