# Peer review of "Engineered Swift Equilibration of brownian particles: consequences of hydrodynamic coupling"

_SciPost Physics, doi:SciPost Phys. 9, 064 (2020)_

## Round 1 · Referee Report · Ramón Castañeda-Priego (Referee 1) · 2020-6-1

Strengths

  1. Careful and systematic (experimental and theoretical) analysis of the Engineered Swift Equilibration (ESE) of two hydrodynamically coupled colloids in optical traps.

  2. Design of an ESE protocol that does not depend on the hydrodynamic coupling.

  3. The ESE protocol can be easily extended to more complicated situations and/or configurations.

Report

Authors report on a systematic (experimental and theoretical) analysis of the Engineered Swift Equilibration (ESE) of two hydrodynamically coupled colloids individually trapped in optical tweezers. They designed an ESE protocol that does not depend on the hydrodynamic coupling. In particular, they found that the presence of the second particle slightly affects the response of the first one; the effect is explained in terms of the hydrodynamic correlations between colloids.

In my opinion, the discussion based on the hydrodynamic coupling is clear and convincing, and the results are interesting and deserve to be published. Furthermore, the manuscript is well written and organized. However, authors should carefully review and correct the English; there are minor typos in the text, prior to the publication of the manuscript. Figures can also be improved.

Requested changes

  1. English should be properly reviewed and corrected; there are minor typos in the text.

  2. Figures can be drastically improved to better understand their content.

---

## Round 1 · Referee Report · Anonymous (Referee 2) · 2020-7-22

Strengths

1) Very readable review of previous work. 2) Clear presentation of the new experiment and its theoretical analysis.

Weaknesses

1) English could be somewhat improved. Examples: statements like "a sum of sinus of different frequencies ...", "Besides the experimental results ..." should be rewritten, "explicit" is rarely used as a verb etc. 2) In Fig. 2 one needs to compare two panels; it would be nice if both used the same independent variable, either dimensionless time s or time in physical units.

Report

The paper presents an extension of the previously developed Engineered Swift Equilibration (ESE) protocol to two hydrodynamically coupled Brownian particles. The experimental results agree well with the theory.

The paper is interesting and clearly written. I recommend that it is accepted after the authors consider referees' comments.

Requested changes

The authors are encouraged to address weaknesses mentioned above.

---

## Round 2 · Author Response

We thank the referees for their positive reviews of our submission. In this minor revision, we address the two points that were raised by both of them: English has been reviewed, and we did our best to improve the figures .

---

## Round 2 · List of Changes

• We proofread the text of the article to improve the English, with an increased attention to the specific points mentioned by referee 2.

  • All of the figures were reworked with the goal of improving their readability. To this aim, a new notation $\sigma_n$ has been introduced in the main text, figures and their captions to better explain the main quantity we are plotting. This normalized variance of the first particle $\sigma_n=(\sigma(t)-\sigma_{f})/(\sigma_{i}-\sigma_{f})$ describes the evolution of the variance of particle under control, normalised so that $\sigma_n=0$ at initial time, and $\sigma_n=1$ at the end of the protocol.

  • The left panel of figure 2 has been changed according to the suggestion of referee 2, to use the same unit for the time variable on the two panels of this figure.

---

## Editorial Decision

published